# Human IgA Monoclonal Antibodies That Neutralize Poliovirus, Produced by Hybridomas and Recombinant Expression

**DOI:** 10.3390/antib9010005

**Published:** 2020-02-28

**Authors:** Rama Devudu Puligedda, Vladimir Vigdorovich, Diana Kouiavskaia, Chandana Devi Kattala, Jiang-yang Zhao, Fetweh H. Al-Saleem, Konstantin Chumakov, D. Noah Sather, Scott K. Dessain

**Affiliations:** 1Lankenau Institute for Medical Research, Wynnewood, PA 19096, USA; PuligeddaR@MLHS.ORG (R.D.P.); KattalaC@MLHS.ORG (C.D.K.); Al-SaleemF@MLHS.ORG (F.H.A.-S.); 2Center for Global Infectious Disease Research, Seattle Children’s Research Institute, Seattle, WA 98109, USA; vladimir.vigdorovich@seattlechildrens.org (V.V.); noah.sather@seattlechildrens.org (D.N.S.); 3Center for Biologics Evaluation and Research, Food and Drug Administration, Silver Spring, MD 20993, USA; diana.kouiavskaia@fda.hhs.gov (D.K.); konstantin.chumakov@fda.hhs.gov (K.C.); 4ATGC, Inc., Wynnewood, PA 19096, USA; jzhao@atgcinc.com

**Keywords:** IgA, monoclonal antibodies, human antibodies, poliovirus, circulating vaccine derived polioviruses, cVDPV, On-Cell mAb Screening, OCMS™, hybridoma methods

## Abstract

Poliovirus (PV)-specific intestinal IgAs are important for cessation of PV shedding in the gastrointestinal tract following an acute infection with wild type or vaccine-derived PV strains. We sought to produce IgA monoclonal antibodies (mAbs) with PV neutralizing activity. We first performed de novo IgA discovery from primary human B cells using a hybridoma method that allows assessment of mAb binding and expression on the hybridoma surface: On-Cell mAb Screening (OCMS™). Six IgA1 mAbs were cloned by this method; three potently neutralized type 3 Sabin and wt PV strains. The hybridoma mAbs were heterogeneous, expressed in monomeric, dimeric, and aberrant forms. We also used recombinant methods to convert two high-potency anti-PV IgG mAbs into dimeric IgA1 and IgA2 mAbs. Isotype switching did not substantially change their neutralization activities. To purify the recombinant mAbs, Protein L binding was used, and one of the mAbs required a single amino acid substitution in its κ LC in order to enable protein L binding. Lastly, we used OCMS to assess IgA expression on the surface of hybridomas and transiently transfected, adherent cells. These studies have generated potent anti-PV IgA mAbs, for use in animal models, as well as additional tools for the discovery and production of human IgA mAbs.

## 1. Introduction

The oral attenuated poliovirus vaccine (OPV) is the most effective vaccine for eradicating wild Polio Virus (PV). Yet, OPV can give rise to circulating Vaccine Derived Polio Viruses (cVDPV), which have genetically reverted to and are clinically indistinguishable from wild PV. For this reason, the Global Polio Eradication Initiative (GPEI) has planned a staged substitution of OPV with the inactivated poliovirus vaccine (IPV). Because type 2 wild PV has been eradicated, the first phase of this transition involved a switch to a bivalent OPV (types 1, 3) combined with a trivalent IPV boost [1]. An important challenge to this strategy is that IPV does not provide definitive mucosal immunity. As a result, cVDPVs can infect the intestines of IPV-immune individuals, be shed in the stool, and pass asymptomatically through IPV-immune populations [2,3]. When these viruses encounter a non-immune population, they can cause outbreaks of poliomyelitis, and the frequency of type 2 cVDPV outbreaks has been increasing since the April 2016 transition. Eighteen outbreaks were detected in Central and Western Africa from January to June 2019, and it is now considered that cVDPV2 has established endemicity in large parts of Africa and Asia [4].

Definitive control of cVDPVs should eventually be possible through the use of genetically stabilized OPV strains [5]. In the meantime, the National Research Council has recommended that least two mechanistically distinct PV drugs be developed as an additional means of responding to cVDPV outbreaks [6]. For example, the small molecule Pocapavir shortened the period of OPV shedding in a randomized clinical trial, and resistant PV strains that arose during that trial were fully susceptible to human mAbs [7,8]. We have been studying human mAbs that have broad anti-PV neutralizing activity and could be administered in combination with small molecules [8]. However, our previous mAbs were all IgGs, whereas IgAs are the primary immoglobulins in mucosal site and are essential to PV mucosal immunity. Cessation of OPV shedding in vaccinated infants correlates with development of an intestinal, PV-neutralizing IgA response [9]. In addition, orally administered IgA (human colostrum) reduced PV shedding in an individual who chronically sheds VDPVs due to an inborn immunodeficiency [10].

As a central component of mucosal immunity, IgAs may protect against some viral infections, regulate the contents of the microbiome, and modulate the risk of atopic diseases and autoimmunity [11,12,13,14,15,16]. Intestinal IgA antibodies are expressed primarily by plasma cells in the lamina propria [17,18]. They are transported to mucosal surfaces as secretory IgAs (sIgA), in the form of dimeric or tetrameric IgA molecules complexed with the immunoglobulin J chain and the Secretory Component (SC), a fragment of the polymeric IgA receptor that is acquired during transfer [17,18]. The intestinal IgA response arises primarily in Peyer’s patches and is distinct from the systemic IgA response, although the cells are clonally related and mucosal homing IgA+ pre-plasma cells can be found in peripheral blood [19,20,21]. Human IgA mAbs have been efficiently isolated from peripheral blood B cells using recombinant and cell-based methods [13,21,22,23,24], while technologies for producing IgA mAbs in plants and mammalian cells continue to improve [25,26,27,28,29,30].

The objective of this study was to produce human IgAs with PV neutralizing activity for evaluation in pre-clinical models of PV infection. We typically use a hybridoma method for cloning human IgM and IgG mAbs from memory B cells [31]. In this study, we adapted this method for cloning human IgAs. In addition, we recently introduced an improved paradigm for screening hybridoma mAbs, On-Cell mAb Screening (OCMS™), in which mAbs are tested on the surface of the cells that produce them. OCMS simplifies the hybridoma process for cloning human mAbs because it uses soluble antigens, does not require that hybridoma supernatants be separated from the hybridomas for screening, and eliminates the need for limiting dilution cloning. Here, we tested the ability of OCMS to assess hybridoma-expressed, human IgAs for PV binding. We used recombinant expression methods to convert PV-neutralizing human IgG mAbs to dimeric IgA1 and IgA2 molecules, and then observed that OCMS can be used to confirm IgA production by transiently transfected cells. These studies establish novel methods for cloning and expressing human, antigen-specific IgAs, while providing PV-neutralizing human IgAs for therapeutic modeling.

## 2. Materials and Methods

### 2.1. Volunteer Blood Donors

Peripheral blood mononuclear cells (PBMCs) were obtained from two PV-exposed individuals, donor P3 (age 30–35 years) and donor P6 (age > 60). Donor P3 formerly lived in a PV endemic country and was exposed to multiple doses of OPV, whereas donor P6 had a possible wild PV infection, as well as multiple lifetime exposures to OPV and IPV. They both received a dose of IPV eight days prior to blood sampling. The P3 and P6 PBMCs used in this study were aliquots of previously described samples [8,32]. Work with human blood cells was performed with informed consent, under protocols approved by the Main Line Hospitals Institutional Review Board.

### 2.2. B Cell Cultures and Hybridoma Creation

Hybridoma methods were performed with cultured PBMCs as described previously, with minor modifications [32]. PBMCs were isolated using ficoll-Paque (17-1440-02; GE Healthcare, Chicago, IL, USA) density gradient centrifugation and cryopreserved. CD27+ cells were MACS-isolated using CD27 microbeads (130-051-601; Miltenyi Biotec, Cambridge, MA, USA). CD27+ cells (2 × 10^5^ cells/well) were plated in a 24-well plate and cultured for eight days in advanced RPMI supplemented with 10% FBS, 100 IU/mL penicillin, 50 μg/mL streptomycin, 5 ng/mL human IL-2, 50 ng/mL human IL-10 (Peprotech, Rocky Hill, NJ, USA), 10% CHO cell conditioned medium containing Ultra CD40L (Multimeric Biotherapeutics, La Jolla, CA, USA), 1 μg/mL CpG (ODN2006, Invivogen, San Diego, CA, USA), 100 ng/mL rhBAFF, 1 μg/mL cyclosporin, 0.5 ng/mL rhTGF-β and 100 ng/mL rhAPRIL (R&D Systems, Minneapolis, MN, USA). On day 8, cells were harvested and fused with the LCX OCMS fusion partner cell line by electrofusion [33]. Positive wells were subjected to three rounds of single cell cloning to establish stable hybridomas secreting anti-PV IgA mAbs. For scale up, hybridoma clones were adapted to 5% Ultra Low IgG FBS (Thermo Fisher Scientific, Waltham, MA, USA) and purified with columns packed with Pierce Jacalin Agarose (20395; Thermo Fisher) following the manufacturer’s recommendations.

### 2.3. Immunocapture ELISAs

Poliovirus specific IgA clones were identified by immunocapture ELISA. Corning Easy-Wash 96-well ELISA plates (3369; Thermo Fisher) were coated with 1:1000 diluted rabbit anti-human IgA mAb (ab193189; Abcam, Cambridge, MA, USA) in PBS and incubated overnight at 4 °C. The following day, plates were washed three times with PBST (PBS with 0.05% Tween 20), then blocked with blocking buffer (PBS with 2% BSA) at 37 °C for 1 h. Three PBST washes at room temp were performed between the following steps. (a) Cell culture supernatant (100 μL/well) was added and incubated at 37 °C for 1.5 h. (b) 50 μL/well Sabin PV3 (1:50 dilution of a 10^9^ (Cell Culture Infectious Doses) CCID_50_/mL stock) was added and incubated at 37 °C for 1.5 h. (c) 1 μg/mL type 3 specific anti-PV IgG mAb, 6B5 (100 μL/well in PBST) was added and incubated at 37 °C for 1 h. (d) Anti-human IgG Fc specific HRP conjugated mAb (9040–05; Southern Biotech, Birmingham, AL, USA) at 1:1500 dilution in PBST was incubated at 37 °C for 1 h. After three final washes, plates were incubated with OPD substrate (P8287; SigmaAldrich, St. Louis, MO, USA) (100 μL/well) for 15 min at 22 °C. The reaction was stopped with 100 μL/well 1N HCL. Optical density (OD490) was read with a Biotek Synergy II microplate reader (BioTek Instruments, Winooski, VT, USA).

IgA isotyping was performed as above with the following secondary antibodies (all from SouthernBiotech, Birmingham, AL and used at 1:1000 dilution): goat anti-human lambda HRP (2070–05), goat anti-human κ HRP (2060–05), mouse anti-human IgA1 HRP (9130–05), and mouse anti-human IgA2 HRP (9140–05). IgA binding to Sabin PV was tested by immunocapture ELISA as above with the following modifications. After coating with rabbit anti-human IgA mAb and blocking with blocking buffer, plates were incubated with 5 μg/mL purified anti-PV IgAs or a non-specific IgG mAb (6A) at 37 °C for 1.5 h. The indicated Sabin PV strains were next added (1:50 dilution of 10^9^ CCID_50_/mL stocks) and incubated at 37 °C for 1.5 h. After three PBST washes, PV was detected with 1 μg/mL anti-PV IgGs (the 2F7 mAb for type 1 and type 2 PV and 6B5 mAb for type 3 PV), followed by the anti-human IgG Fc specific HRP conjugated mAb. All samples were analyzed in triplicate.

### 2.4. OCMS Analysis of mAbs Expressed by Human Hybridomas

Capture of human IgA and type 3 Sabin PV on the surface of OCMS hybridomas was assessed by fluorescence microscopy. For the OCMS hybridomas, 10^5^ cells were plated on round Corning BioCoat 12 mm #1 German Glass coverslips (354087; Thermo Fisher) in 24-well plates and incubated overnight with 1 μg/mL rabbit anti-human IgA mAb (ab193189; Abcam) in culture medium at 37 °C, 5% CO_2_. The cells were washed and incubated with 20 µL biotinylated Sabin type 3 PV (8 × 10^5^ PFU) in PBS 1% BSA, and PV binding was detected with 1:200 Alexa Fluor 488 Streptavidin (016-540-084; Jackson ImmunoResearch, West Grove, PA, USA). PV was biotinylated with the EZ-Link Sulfo-NHS-LC-Biotinylation Kit (21435; Thermo Fisher). Human IgA binding to the PV OCMS hybridomas was detected with 1:200 Alexa Fluor 647 conjugated goat F(ab’)2 anti-human IgA (2052-31; SouthernBiotech). Cells were processed for confocal microscopy by washing three times with PBS 1% BSA and fixing with 4% paraformaldehyde (PFA) in PBS for 15 min at RT. After fixation, cells were washed three times with PBS 1% BSA, once with PBS, and once with dH_2_O. The coverslips were mounted with ProLong Gold Antifade reagent with DAPI (P36935; Thermo Fisher). Slides were imaged with a C2+ Nikon confocal microscope with 63x/1.3 NA oil objective; images were analyzed with ImageJ software [34]. Hybridoma populations in 96-well plates (following HAT selection) were handled similarly, except that the plates were imaged with a PerkinElmer Operetta high content imaging system.

### 2.5. Poliovirus Microneutralization Test

PV-neutralizing mAb titers were determined in a microneutralization test [17,23]. The mAbs were diluted to 10 μg/mL in maintenance medium (DMEM with 2% FBS and 1% antibiotic/antimycotic solution; Thermo Fisher) and sterilized by filtration through Corning Spin-X columns (Thermo Fisher). Twofold serial dilutions of the mAbs (in duplicates or triplicates) were incubated for 3 h at 36 °C with 100 CCID_50_ of the respective PV strain in an atmosphere of 5% CO_2_. After the incubation, 1 × 10^4^ HEp-2c cells were added to the wells. The plates were incubated for 10 days at 36 °C, 5% CO_2_ and evaluated microscopically. Neutralizing antibody titers were calculated using the Kärber formula and normalized to a 1 mg/mL solution [35].

### 2.6. FPLC Gel Filtration Analysis and Immunoblotting

Purified anti-PV IgA mAbs from hybridomas were dialyzed against PBS and fractionated by size exclusion chromatography (SEC) on a Superose 6 HR 10/300 column (17-0537-01; GE Healthcare) connected to an ÄKTA FPLC system (GE Healthcare). The column was calibrated with a wide range (143.7 to 443 kDa) of molecular weight markers (MWGF1000; Sigma Aldrich). Approximately 150 μg antibody in 100 μL was loaded onto the column at a flow rate of 0.5 mL/min in PBS and two 5 mL fractions containing dimeric IgA (fraction 1) and monomeric IgA (fraction 2) were collected. IgA mAbs from recombinant expression were analyzed by gel filtration of 20 µg mAb samples with a Superdex 200 Increase 10/300 GL column (28990944; GE Healthcare) and Gel Filtration Standards (1511901; Bio-Rad, Hercules, CA, USA).

For J-chain analysis of the hybridoma-expressed IgAs, FPLC fractionated human anti-PV IgA (10 μg) samples were resolved by reducing SDS-PAGE on a 4–12% (*w*/*v*) polyacrylamide gel with the Criterion system (Bio-Rad) and transferred to a nitrocellulose membrane (88024; Thermo Fisher). The membrane was blocked by incubation in 5% non-fat dry milk in TBS plus 0.05% Tween 20 (TBST) overnight at 4 °C. Subsequent incubations were all performed in TBST. After three washes with TBST, the blocked membrane was incubated with HRP conjugated goat anti-human J chain of dimeric IgA (MBS571594; MyBioSource, San Diego, CA) at 1:1000 dilution for 1 h, washed three times for 5 min, then incubated with HyGlo Quick Spray Chemiluminescent HRP detection solution (E2410; Thomas Scientific, Swedesboro, NJ, USA) for 1 min, and the substrate was detected with a light-sensitive camera (Chemidoc; Bio-Rad).

### 2.7. Expression Constructs for Recombinant IgA Expression

Constructs encoding antibody heavy and light chains were built from gBlock DNA fragments (Integrated DNA Technologies, Redwood City, CA, USA) and assembled using NEBuilder HiFi DNA Assembly Master Mix (New England Biolabs, Ipswich, MA, USA) [36]. The κ-acceptor vector was built using a murine κ chain leader sequence (Figure 1), which was separated by a NotI restriction site from the human κ-chain constant region (IGKC*01) in the pcDNA3.4 backbone plasmid (Invitrogen, Carlsbad, CA, USA). Two alpha-chain acceptor vectors were built with a murine heavy chain leader separated by a NotI site from the human alpha-chain constant regions (IGHA1*01 and IGHA2*01, each resulting in its own acceptor vector) in pcDNA3.4. Similarly, a gamma-chain acceptor vector was generated using the murine heavy chain leader with a human gamma-chain constant region (IGHG1*02). A gBlock encoding human J-chain (hIgJ) sequence (GenBank:XM_011531926) was assembled with pcDNA3.4. Acceptor vectors were linearized by digestion with NotI (New England Biolabs) and joined to variable-gene segments from anti-poliovirus human mAbs 1G1 (Genbank Accession # pending) and 2F7 (Genbank Accession # pending), resulting in 1G1.hIgG1, 1G1.hIgA1, 1G1.hIgA2, 1G1.hIgK, 2F7.hIgA1, 2F7.hIgA2, and 2F7.hIgK plasmids.

### 2.8. Recombinant mAb Expression and Purification

Protein production was performed using transient transfection of HEK293F cells, as previously described [36]. HEK293F cells were co-transfected as follows: hIgG1/hIgK plasmids were used at 0.5/0.5 ratio, whereas hIgA1/hIgK/hIgJ or hIgA2/hIgK/hIgJ plasmids were used at 0.25/0.25/0.5 ratio. Following 5 days of culture, conditioned medium was harvested by centrifugation, supplemented by the addition of NaN_3_ (0.02% final concentration), and NaCl (+350 mM, final concentration). Protein was then captured using Pierce Protein A Plus (Fisher Scientific), for IgG purification, or Pierce Protein L Plus Agarose (Fisher Scientific), for dIgA purification, washed with HBS-E-hs buffer (10 mM HEPES, pH 7, 300 mM NaCl, 2 mM EDTA), and eluted in 0.1 mM glycine, pH 2.7 (fractions were immediately pH-neutralized using 1 M Na_2_HPO_3_). Protein-containing fractions were pooled and buffer-exchanged against HBS-E (10 mM HEPES, pH 7, 150 mM NaCl, 2 mM EDTA) by ultrafiltration.

### 2.9. Generation and Screening of κ-Chain Mutants

To facilitate the binding of 1G1 κ light chain IgG to Protein L affinity column, we generated two mutant constructs, namely Mutant 1 and Mutant 2. Mutant 1 had ten mutations in the κ-chain sequence (P12S, T14S, P18R, S20T, L42Q, Q50K, G56A, K79T, V83L, and G89A), whereas Mutant 2 had five mutations (P12S, P18R, L42Q, Q50K, K79T). These mutations were synthesized as gBlock DNA fragments (IDT) and cloned into the κ-acceptor vector, as described for the wild-type 1G1 κ-chain sequence. Further mutants were generated by PCR-amplification of fragments including the target mutations derived from Mutant 2 or wild-type (wt) 1G1 sequences. These PCR fragments were then used for Mutant 2/wt swapped cloning into an empty pcDNA3.4 vector using NEBuilder HiFi DNA Assembly Master Mix (New England Biolabs), resulting in Mutant3 (P12S), Mutant 4 (P12S/P18R), and Mutant 5 (L42Q/Q50K/K79T).

Following sequence verification, HEK293F suspension cells were cotransfected with plasmids encoding 1G1.hIgG1 heavy chain and each of the mutant 1G1 κ-light chain constructs using 293-Free transfection reagent (721-81; Millipore Sigma, Burlington, MA, USA). Following five days of culture, supernatants were clarified by centrifugation and supplemented with 0.02% NaN3 and 350 mM NaCl. Samples were analyzed by Western blot and the IgG was detected with HRP conjugated goat anti-human IgG (H+L) secondary antibody (31410; Invitrogen). To test the binding of IgG to protein L agarose resin, 1.5-mL each culture supernatant sample was incubated for 45 min with 50 µL PBS-equilibrated Pierce Protein L Plus Agarose (20520; Fisher Scientific). After incubation, the resin was collected by centrifugation and washed twice with HBS-E buffer, then the samples were boiled with gel loading buffer and analyzed by SDS-PAGE.

### 2.10. Binding of Recombinant IgA to 293T OCMS Cells

293T OCMS and 293T cells were plated at 3 × 10^4^ cells/well on German Glass Coverslips in 24-well plates in DMEM with 10% FBS. The following day, cells were transfected with 1 µg each recombinant 1G1 IgA heavy and light chain expression plasmids with X-tremeGENE 9 DNA transfection reagent (06 365 787 001, Sigma Aldrich). One day after transfection, the culture medium was replaced with 1 mL fresh medium with 1 μg/mL rabbit anti-human IgA mAb (ab193189; Abcam) and the cells were incubated for 24 h at 37 °C. The cells were washed 3 times with PBS 1% BSA, co-incubated for 1 h with 1:200 Alexa Fluor 647 conjugated goat F(ab’)2 anti-human IgA (2052-31; SouthernBiotech) and 1:200 Alexa Fluor 488 conjugated AffiniPure F(ab’)2 fragment goat anti-rabbit IgG (H+L) (111-546-144; Jackson ImmunoResearch) to detect the expression of IgA and OCMS respectively. The cells were then analyzed by confocal microscopy (see Section 2.4).

## 3. Results

### 3.1. Human Monoclonal IgA Antibodies Specific for Poliovirus the Other Four IgAs Demonstrated

To clone human IgA mAbs specific for PV, we used peripheral blood from previously described donors, P3 and P6. Donor P3 was exposed to multiple doses of OPV and donor P6 had received multiple doses of OPV and IPV. They both received an IPV boost 8 days prior to blood sampling [32]. We cultured CD27+ peripheral blood mononuclear cells (PBMCs) with Ultra CD40L, IL-2, IL-10, CpG oligonucleotide ODN2006, TGF-β, APRIL and BAFF [32,37,38,39,40] for 8 days and then fused the cells to the LCX OCMS™ cell line [33]. We performed two cell fusions with PBMCs from these donors and plated the cells in ten 96-well plates (5 plates from each donor). After 14 days in HAT selection, the hybridomas were screened for IgA reactivity with type 3 Sabin PV by immunocapture ELISA. We identified a total of six clones, one clone (1E4) from donor P3 and five clones (1A12, 1H10, 2A6, 2D10 and 5F10) from donor P6. We expanded these to a 24-well plate. After 4 days of culture, we plated a sample from each well on cover slips and used the OCMS method (Section 3.2) to confirm that all of the wells contained IgA-expressing cells (data not shown). We subjected each of the populations to three rounds of single-cell cloning to establish stable hybridomas secreting human IgA mAbs specific for PV. Isotyping analysis revealed that all mAbs were IgA1 with lambda light chains. Hybridoma clones were adapted to ultra-low IgG medium and purified with Pierce Jacalin Agarose. Yields were approximately 4 mg per 500 mL culture, similar to what we previously observed with IgG mAbs.

The PV binding of IgA mAbs was assessed by immunocapture ELISA, in which IgA mAbs were captured by rabbit anti-human IgA and their binding to monovalent Sabin PV was detected by type specific anti-PV IgGs (2F7 for types 1, 2 and 6B5 for type 3). The mAbs all exhibited measurable binding to type 3 PV, and two of the mAbs (1E4 and 1A12) showed multi strain reactivity. 1E4 bound to both type 2 and type 3 PV, whereas 1A12 bound to all three PV serotypes (Figure 2). IgG mAbs with similar multi-type specific PV binding have been observed with these donors previously [32].

### 3.2. OCMS Testing of Hybridomas Secreting Human Anti-PV IgA mAbs

The LCX OCMS cell line, which we used to form the hybridomas, expresses a tandem scFv “Anchor”, which is specific for rabbit IgG [33]. Anchor expression is maintained in hybridomas following cell fusion. Thus, mAbs expressed by OCMS-enabled hybridoma libraries can be screened for antigen binding specificity by incubating the cells with a rabbit anti-human Ig secondary antibody, the “Linker”, which adheres to the cells and captures secreted Igs, followed by fluorescent antigen. The Ig capture and display reaction is specific because the binding conditions include excess Linker, which competitively inhibits Ig complexes from binding to cells that did not secrete the IgGs.

We used the OCMS method to assess the hybridomas for secretion of PV-specific IgAs, testing PV binding to the cells after an overnight period of mAb capture (Figure 3). The IgA-expressing OCMS hybridomas were cultured overnight with the rabbit anti-human IgA mAb. The IgG-expressing OCMS hybridoma, 8C5, was used as a negative control [33]. The next day, the cells were washed and incubated one hour with biotinylated type 3 PV. After washing, Alexa Fluor 488 streptavidin was added to detect bound PV (green), and Alexa Fluor 647 conjugated goat F(ab’)2 anti-human IgA was added to detect IgA (magenta). All of the PV hybridoma cultures had cells positive for IgA, and many of the IgA signals co-localized with PV, whereas the 8C5 hybridoma did not bind IgA or PV.

### 3.3. Gel Filtration Analysis of Hybridoma Expressed IgAs

We assessed the IgAs by gel filtration with a Superose 6 HR 10/300 column (Figure 4a). We collected the eluates in two fractions (F1, F2), then analyzed these for J chains by Western blotting (Figure 4b). Each of the IgA mAbs demonstrated a peak at ~33 min, which matched the molecular weight marker, alcohol dehydrogenase (150 kDa) and was only slightly slower than the two monomeric IgGs tested (8C5 and 1G1). Western blotting indicated that the F2 fraction, which contained these peaks, did not contain the J chain (2D10 is an exception). Together, these data suggest that all of the hybridomas express monomeric IgA1, and that 2D10 and 5F10 comprise entirely monomeric species.

The other four IgAs demonstrated complex patterns. 1A12 and 1H10 both had peaks at ~27 min in the F1 fraction, for an estimated size of 640 kDa, as well as J chain expression. This suggests a dimeric form, as our recombinant dimers ran similarly in gel filtration (see Section 3.6) and non-reducing SDS:PAGE (data not shown). The planar structure of dimeric IgAs likely accounts for its discordance with the molecular weight standards [41]. 2A6 also had the monomer and dimer peaks, but also a third, middle peak. Because the middle and slowest peaks both fall within F2, and no J chain was seen in the 2A6 F2 fraction, it is likely that the middle peak is a variant monomer species. In 2D10, two superimposed peaks ran in the F1 fraction, which was associated with a J chain, suggesting a 2D10 monomer population linked to a J chain, or a slow-moving dimer. Thus, human IgAs expressed by hybridomas can exist in heterogeneous monomeric and multimeric forms.

### 3.4. Poliovirus Neutralizing Studies with Hybridoma-Expressed IgA mAbs

We tested the purified mAbs for neutralization of type 3 Sabin and wild PV strains using the microneutralization test (Table 1). Three of the mAbs were highly active (1H10, 2A6, and 2D10), with titers ranging between 86,100 and 688,900 dilution of a 1 mg/mL mAb solution. These high activities correlated with a complex gel filtration profile with J chain expression (Figure 4). In contrast, 1A12 also had multimeric species, but minimal neutralizing activity. Low neutralizing activity was seen with both of the monomeric IgAs (1E4 and 5F10). These levels of neutralization are similar to those measured previously with IgG mAbs from these same individuals, in particular the type 3-specific IgGs 6D11, 7A1, and 7E2 [8]. In addition, 1A12 resembles the LX_10C6 IgG, which binds all three serotypes but has minimal neutralizing activity [32].

### 3.5. Expression of Recombinant Dimeric IgAs

We selected two human IgG mAbs for conversion into IgA mAbs: 2F7 and 1G1. Both contain κ light chains, which specifically bind to Protein L, to assist in purification. 2F7 is highly active against type 1 PV, but also neutralizes type 2 PV (Table 2) [32]; 1G1 neutralizes only type 3 PV (Table 2). We used gBlock DNA fragments to assemble recombinant genes that encode 2F7 and 1G1 heavy chains as IgG1, IgA1, and IgA2 isotype molecules, as well as fragments encoding 2F7 and 1G1 κ light chains and an immunoglobulin J-chain [36].

In our initial expression tests, we observed that the recombinant 1G1 mAb did not bind to Protein L (data not shown). Examination of the amino acid sequence of the 1G1 κ chain indicated that it resembles the VκII subtype, which does not bind to Protein L [42]. We created a series of variant 1G1 κ genes, mutated in their framework regions (FR), based on a primary sequence comparison to the non-VκII subtypes (Figure 5) and a 2.7 Å crystal structure of a Protein L:Fab complex [43]. Mutant 1 introduced a serine at position 12 (P12S) and an arginine at position 18 (P18R), both of which are implicated in Protein L binding, as well as eight additional changes (T14S, S20T, L42Q, Q50K, G56A, K79T, V83L, G89A). In contrast, Mutant 2 targeted a subset of five of these residues: P12S, P18R, L42Q, Q50K and an FR2 threonine (K79T). Both of these expressed well and bound Protein L (Figure 5B,C). Mutant 3 contained only the P12S mutation, whereas Mutant 4 contained both P12S and P18R. Both of these also bound Protein L. Lastly, in Mutant 5, mutation of three charged residues in FR2 and FR3 (L42Q, Q50K, K79T) did not enable Protein L binding. Some variability was noted in the expression levels of the unpurified IgAs (Figure 5B), but this did not substantially affect the yields of purified IgAs (Figure 5C). These data demonstrate the importance of the P12 residue to disruption of Protein L binding, which was previously proposed [43], and are consistent with the finding that a single S12P mutation in a human κ LC prevented binding to Protein L [44]. Thus, we selected the 1G1 P12S κ mutant (Mutant 3) for recombinant expression.

### 3.6. Production and Purification and Testing of Dimeric IgAs

Recombinant IgA1 and IgA2 mAbs were produced by transient transfection of the relevant plasmids into HEK293F cells. Recombinant IgG mAbs were also made as positive controls for activity. The mAbs were purified by Protein L affinity chromatography. Yields per 1 L culture volume were: 1G1 IgA1, 10.5 mg; 1G1 IgA2, 13.8 mg; 2F7 IgA1, 14.2 mg; 2F7 IgA2, 10.4 mg. Gel filtration and SDS:PAGE analyses are shown in Figure 6.

### 3.7. Neutralization of PV by Recombinant mAbs

We tested the neutralization ability of the recombinant 2F7 and 1G1 IgA1 and IgA2 mAbs, comparing them to the activities seen with the original hybridoma-expressed IgGs and the recombinant IgGs. Titers for the recombinant 2F7 mAbs were comparable to the hybridoma-derived 2F7 IgG (within ~2-fold) against type 1 Sabin and wt viruses. The only exception was the IgA2 mAb, which had <1/5 activity against type 1 wt PV, compared to the 2F7 IgG. Neutralization titers of the the 1G1 IgA1 and IgA2 mAbs were essentially the same as what was measured with the hybridoma-derived 1G1 IgG (Table 2). Activity of the recombinant IgG against the Sabin type 3 PV was three-fold lower compared to hybridoma-expressed 1G1, even though its activity against type 3 wt was somewhat higher. These disparities may be related to the single amino acid difference in their κ light chains. Overall, these data demonstrate that the neutralization ability of the IgGs could be preserved by conversion to recombinant IgA mAbs.

### 3.8. Assessment of Recombinant IgA Expression in Transiently Transfected Cells Using OCMS

We transfected 293T OCMS and 293T cells with plasmids encoding the Ig heavy chain and light chain of the 1G1 IgA1. Two days later, we incubated the cells overnight with the rabbit anti-human IgA mAb (Linker). After washing, the Linker was detected with Alexa-Fluor 488 anti-rabbit IgG secondary antibody (green) and the captured human IgA with Alexa-Fluor 647 anti-human IgG (magenta) (Figure 7). Human IgA and Linker staining was observed on the outer plasma membrane of the 293T OCMS cells, but not on the transfected 293T cells. Linker and IgA labeling patterns were highly similar, but Linker staining (indicative of stable Anchor expression), was more consistent than IgA labeling, suggesting variability among IgA expression levels, similar to what we observed previously with IgG transfection of these same cells [33]. These results demonstrate that the OCMS method can be used to assess recombinant IgA mAbs expressed by adherent cells.

## 4. Discussion

IgA antibodies are important for overcoming intestinal infections, shaping the microbiome, and preventing autoimmunity, but the therapeutic potential of IgA mAbs is relatively unexplored, in part due to technical challenges [45,46]. In PV infection, an anti-PV mucosal IgA response correlates with resolution of PV shedding from the gastrointestinal tract, and orally administered colostrum was able to transiently reduce PV shedding in a chronically infected individual [9,10]. To explore how IgAs control PV infection and to assess their therapeutic potential, IgA mAbs with potent PV neutralizing activity are required.

Human IgA mAbs have been isolated from memory B cell populations using recombinant methods [13,21,22,23], and IgAs have been produced in plants [26,29,47,48] and mammalian cells [25,28,30,49]. In this study, we produced PV-neutralizing IgA mAbs using two strategies: (1) de novo IgA cloning from primary human B cells using an updated hybridoma method; and (2) isotype switching of existing IgG mAbs, with recombinant expression in transiently transfected HEK293F cells. MAbs produced by both methods have potent PV neutralizing activity and will be useful for in vivo studies.

Hybridoma methods for human mAb cloning are technically straightforward and can be implemented in any laboratory competent in basic cell culture techniques. Hybridomas produce full-length mAbs, without the need for recombinant expression, in quantities sufficient for in vitro and in vivo characterization. As hybridomas are not suitable for production of pharmaceutical-grade mAbs, we used transient transfection of cloned mAb genes to establish proof of concept for the manufacture of 1G1 and 2F7 IgA mAbs. In addition, the OCMS hybridomas described here can be screened using mAb:antigen capture and fluorescent imaging. Adapting the OCMS method to high-throughput fluorescent imaging techniques should simplify the mAb cloning process by eliminating the need for ELISAs and limiting dilution cloning.

We fused a genetically stabilized fusion partner cell line to primary human memory B cells to create hybridomas secreting human mAbs [8,32,50]. We used cryopreserved aliquots of blood samples that were previously used for cloning human anti-PV IgGs, but we altered our pre-fusion B cell culture conditions to include BAFF and April, two factors known to stimulate IgA class switching [39,51]. From two cell fusions, we identified six hybridomas secreting anti-PV IgA mAbs, three of which had neutralizing activity comparable to IgG mAbs previously isolated from these same blood samples. All of the mAbs were IgA1 isotype, and at least three of the hybridomas expressed dimeric IgA associated with J chain expression, as well as monomeric IgA. Neutralizing activity did not correlate with monomeric vs. dimeric structures. These results are consistent with observations that the peripheral blood memory B cell population contains IgA1- and IgA2-expressing cells [52], many of which produce polymeric IgA [53]. Further studies will explore whether J chain expression by hybridomas correlates with polymeric IgA production in vivo or is acquired during the fusion procedure [54].

The fusion partner cell line we used, LCX OCMS, creates hybridomas that can specifically capture and display their antibodies on their outer plasma membranes [33]. The process is initiated when the cells are incubated with a Linker, a rabbit anti-human mAb that binds to the Anchor protein on the cell surface and captures mAbs secreted by the hybridomas. Bound antibodies can be assessed using fluorescent imaging with labeled antigen (PV) and/or fluorescent secondary antibodies. Here, we used OCMS to validate IgA expression and PV binding, which confirmed the genetic stability of the hybridomas at the level of individual cells. We also used OCMS to screen two of the ten 96-well hybridoma fusion plates, examining the fluorescence intensity with a PerkinElmer Operetta. This confirmed one of our positive clones (1E4) but did not find any additional positives. Preliminary tests with a Celigo Imaging Cytometer (Nexcelom Bioscience, Lawrence, MA, USA) suggest that the fluorescent signals produced by OCMS will be sufficient for high-throughput screening applications.

We converted two high-potency anti-PV IgG mAbs into dimeric IgA1 and IgA2 mAbs. We adapted our previous protocols for recombinant mAb expression, to generate isotype switched expression constructs by gene assembly using gBlock DNA fragments [36]. MAbs were expressed by co-transfecting HC, LC, and J chain constructs and purified using Protein L agarose affinity chromatography. Both of the IgG mAbs maintained their neutralization activity against Sabin and wt viruses when they were converted to dimeric IgAs or expressed as recombinant IgGs. These can be tested as dimers or linked to recombinant S component to create secretory IgA [55].

Protein L binds many human κ LC sequences, and incorporating Protein L binding sequences into IgA mAb sequences has been established as a method to facilitate IgA purification [29,48,56]. Both of our selected mAbs possess κ light chains, but the 1G1 LC was a VIIκ type that does not bind Protein L. We therefore created and tested 1G1 κ LC sequences that contained amino acid changes likely to enable binding. Interestingly, a single amino acid change in the first framework region (P12S) was sufficient to confer Protein L binding. The analogous mutation in reverse (S12P) has been previously noted to abrogate Protein L binding by an scFv [44]. Lastly, we demonstrated that the OCMS method can be used to capture and display IgA mAbs on transiently transfected, adherent cells (293T OCMS). This method adds an important capability to the study and production of human IgA mAbs, as it may be used to optimize transfection protocols, assess the antigen binding activities of a series of mAb variants, or confirm the assembly of multimeric IgA complexes.

## 5. Conclusions

Human IgA monoclonal antibodies have been produced that have potent poliovirus neutralizing activity.IgA mAbs were cloned using by two methods: a hybridoma method with human memory B cells and by recombinant expression of isotype-switched immunoglobulin genes.On-Cell mAb Screening (OCMS) was used to characterize IgA expression by hybridomas and transiently transfected cells.These studies expand the toolkit for studying human IgA mAbs.

## 6. Patents

The OCMS method and the LCX OCMS cell line are subjects of a patent application submitted by the Lankenau Institute for Medical Research.

## Figures and Tables

**Figure 1 antibodies-09-00005-f001:**
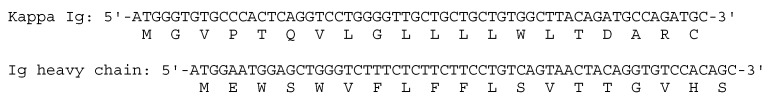
*M. musculus* leader sequences used for recombinant protein expression.

**Figure 2 antibodies-09-00005-f002:**
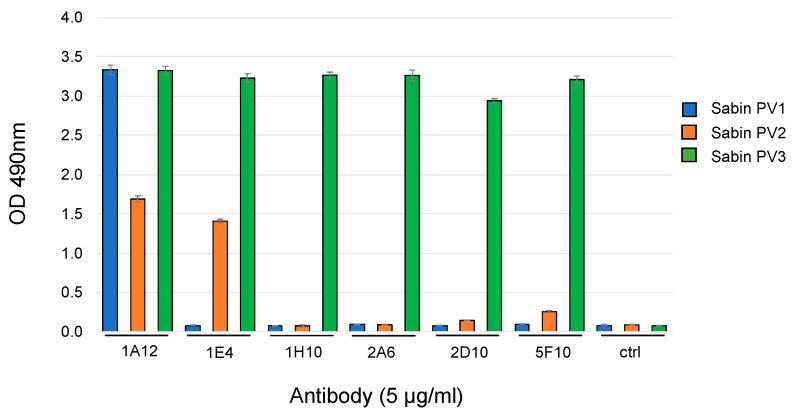
Binding of human IgA mAbs to whole Sabin PV by ELISA. Error bars = S.E.M.

**Figure 3 antibodies-09-00005-f003:**
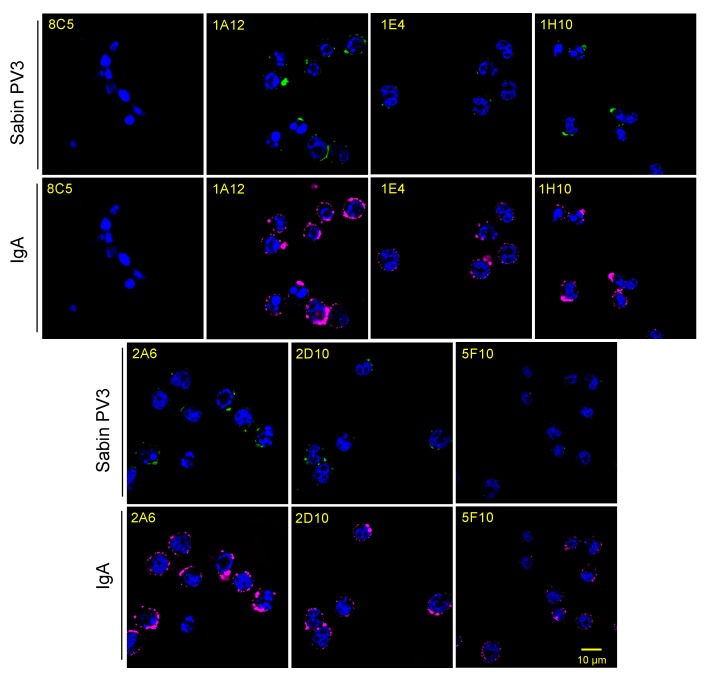
Binding of type 3 PV to OCMS hybridomas secreting anti-PV IgA mAbs. Hybridomas were incubated overnight with the rabbit mAb specific for human IgA and then with biotinylated PV. Human IgA and PV3 binding to the cells were detected with an Alexa-Fluor 647 anti-human IgA secondary and Alexa-Fluor 488 conjugated streptavidin, respectively. Nuclei were stained with DAPI. Scale bar = 10 µm.

**Figure 4 antibodies-09-00005-f004:**
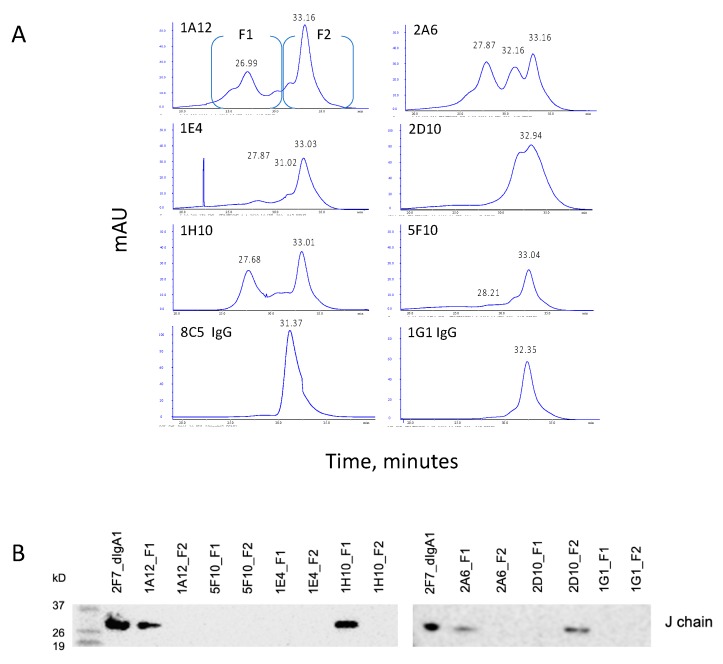
Analysis of the human hybridoma IgA mAbs after Jacalin purification. (**A**) Gel filtration with a Superose 6 HR 10/300 column. Protein peaks for all samples were collected in the two fractions shown (F1, ~27 min; F2, ~33 min). 8C5 IgG and 1G1 IgG are used as monomer controls. (**B**) Western blot analysis of the F1 and F2 fractions for expression of the Ig J chain. Recombinant 2F7 dIgA1 (Section 3.6) was used as a positive control. Molecular weight markers (in kD) are indicated on the left.

**Figure 5 antibodies-09-00005-f005:**
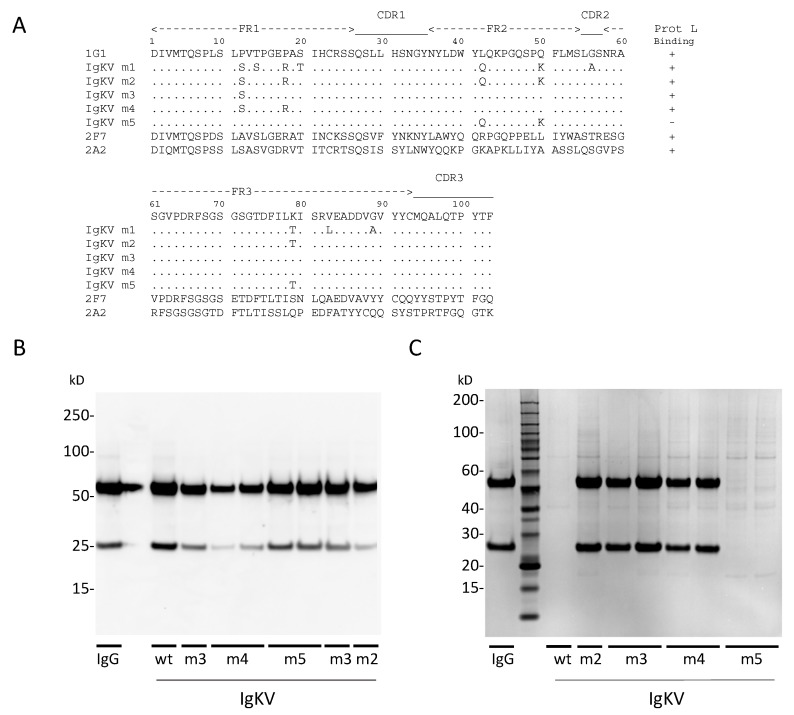
Assessment of human κ light chain variants for binding to Protein L. (**A**) The amino acid sequence of the 1G1 κ light chain (VκII subtype) is shown, as well as the amino acid mutations tested. The framework regions (FR1, FR1, FR3) and complementarity determining regions (CDR1, CDR2, CDR3) are shown above. Also shown are the sequences of 2F7 (VκIV subtype) and 2A2, the VκI light chain used for the reference crystal structure [43]. (**B**) Western blot of IgGs expressed in supernatants of cells co-transfected with κ LC wt or mutant genes as shown in (**A**) or the 1G1 IgG heavy-chain gene. Double lanes for some mutants represent independent transfections. (**C**) Supernatants shown in (**B**) were run through Protein L columns and retained IgGs were analyzed by SDS-PAGE. IgG, purified 1G1 IgG, expressed with P12S κ mutation.

**Figure 6 antibodies-09-00005-f006:**
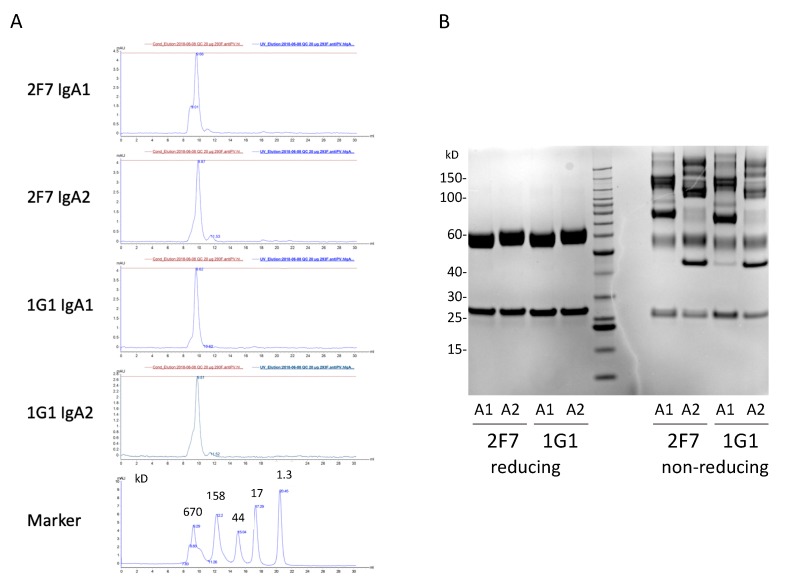
Analysis of the recombinant IgA mAbs after Protein L purification. (**A**) Gel filtration of 20 µg mAb samples with a Superdex 200 Increase 10/300 GL column. Marker, Bio-Rad Gel Filtration Standards. (**B**) Analysis of the purified proteins on reducing (left) and non-reducing (right) SDS-PAGE.

**Figure 7 antibodies-09-00005-f007:**
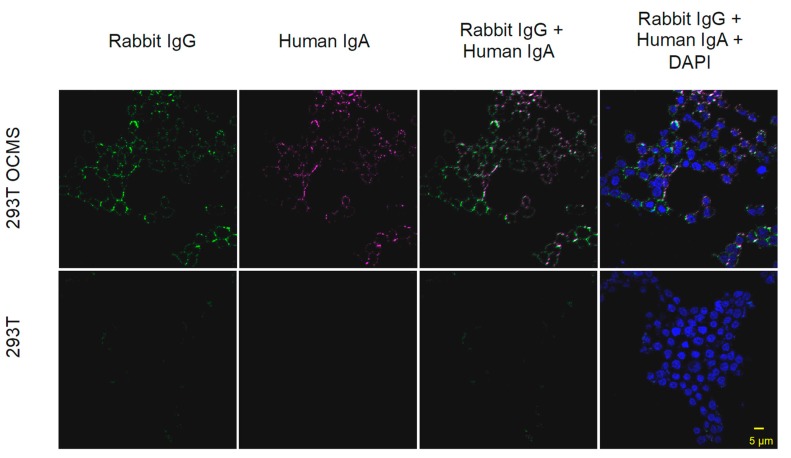
Expression of recombinant IgA in 293T OCMS cells. 293T OCMS (**top row**) and 293T (**bottom row**) cells were transiently transfected with plasmids encoding recombinant 1G1 IgA HC and LC genes. The next day, the cells were incubated overnight with the rabbit anti-human IgA mAb (Linker). Bound Linker was detected with Alexa-Fluor 488 anti-rabbit IgG secondary antibody (green). Captured human IgA was labeled with Alexa-Fluor 647 anti human IgA (magenta). Nuclei were stained with DAPI (blue). Scale bar = 5 µm.

**Table 1 antibodies-09-00005-t001:** Poliovirus neutralizing activities of the hybridoma-expressed IgA mAbs.

Antibody	FPLC	WT 1	Sabin 3	WT 3
1A12	two peaks	x	ND	600
1E4	monomer	x	800	7600
1H10	two peaks	x	102,400	204,800
2A6	three peaks	x	688,900	409,600
2D10	superimposed peaks	x	86,100	102,400
5F10	monomer	x	400	1300

Results are given as the reciprocal dilutions that protected 50% of HEp-2c cells from 100 CCID_50_ of the indicated wt and Sabin PV strains, normalized to 1 mg/mL mAb concentration. Source: H, hybridoma; R, recombinant.

**Table 2 antibodies-09-00005-t002:** Poliovirus neutralizing activities of the recombinant IgA mAbs.

Antibody	Source	Sabin 1	WT 1	Sabin 3	WT 3
2F7_IgG	H	409,600	579,262	x	x
2F7_rIgG	R	237,449	949,797	x	x
2F7_dIgA1	R	402,265	1,137,778	x	x
2F7_dIgA2	R	402,265	100,566	x	x
1G1_IgG	H	x	x	204,800	102,400
1G1_rIgG	R	x	x	72,408	144,815
1G1_dIgA1	R	x	x	289,631	102,400
1G1_dIgA2	R	x	x	204,800	144,815

Results are given as the reciprocal dilutions that protected 50% of HEp-2c cells from 100 CCID_50_ of the indicated wt and Sabin PV strains, normalized to 1 mg/mL mAb concentration. Source: H, hybridoma; R, recombinant.

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
