# Peer review of "Human IgA Monoclonal Antibodies That Neutralize Poliovirus, Produced by Hybridomas and Recombinant Expression"

_2073-4468, 2020, doi:10.3390/antib9010005_

Round 1

Reviewer 1 Report

This manuscript is well written. Only two minor comments,

Result 3.4: How does the mutations on light chain where chosen and how these mutations impact the final yield of the IgAs? A comparision with wildtype control would be a nice addition.  Result 3.5: Please include total yield of 2F7 and 1G1. 

Author Response

Dear Reviewer,

Thank you for your thoughtful review.

"Result 3.4: How does the mutations on light chain where chosen and how these mutations impact the final yield of the IgAs? A comparision with wildtype control would be a nice addition.  Result 3.5: Please include total yield of 2F7 and 1G1."

We significantly expanded the discussion in 3.4.  We also comment on the differences in production between wt and the mutants (Figure 4B), although this is qualitative information, but note further that the final yields of the mutated versions were very similar.  We included yields for all of our preps, including the hybridoma mAbs (all ~4 mg/500 ml), and the 2F7 and 1G1 recombinants.

Reviewer 2 Report

The study is a methodology-based approach to in vitro generation of the new neutralizing IgA-class antibodies against poliovirus. The medical merit of the research is sound and well explained in the introduction. Two separate methods are being used in the study and the results are clearly described. However, in the discussion section only a summary of the results is presented without really discussing the impact of the study. What would increase the value of the manuscript is providing some comparison of the pros and cons of both methods used, especially in relation to other existing methods of generation of IgA mAbs in vitro.

Minor comment:

The affiliation of Dr. Scott K. Dessain is unclear, as it states “a”, and only numbers are used for the affiliation addresses.

Author Response

Dear Reviewer,

Thank you for your thoughtful review.

"However, in the discussion section only a summary of the results is presented without really discussing the impact of the study. What would increase the value of the manuscript is providing some comparison of the pros and cons of both methods used, especially in relation to other existing methods of generation of IgA mAbs in vitro."

We added context to the study starting at line 429, emphasizing the distinctive features of the methods described in this study. We mentioned other methods in the previous paragraph. We hope this is adequate, as most Antibodies readers are familiar with the basics of mAb cloning, and we don't want to imply that other methods (such as single cell cloning and yeast display) are inadequate.